# An Ethics of Needs: Deconstructing Neoliberal Biopolitics and Care Ethics with Derrida and Spivak

Tiina Vaittinen 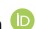

Global Health and Social Policy, Health Sciences, Faculty of Social Sciences, Tampere University, 33100 Tampere, Finland; tiina.vaittinen@tuni.fi

**Abstract:** The body in need of care is the subaltern of the neoliberal epistemic order: it is that which cannot be heard, and that which is muted, partially so even in care ethics. In order to read the writing by which the needy body writes the world, a new ethics must be articulated. Building on Jacques Derrida's philosophy of deconstruction, Gayatri Chakravorty Spivak's notions of subalternity and epistemic violence, critical disability scholarship, and corporeal care theories, in this article I develop an ethics of needs. This is an ethical position that seeks to read the world that care needs write with the relations they enact. The ethics of needs deconstructs the world with a focus on those care needs that are presently responded to with neglect, indifference, or even violence: the absence of care. Specifically, the ethics of needs opens a space—a spacing, an *aporia*—for a more ethical politics of life than neoliberal biopolitics can ever provide, namely, the politics of *life of needs*.

**Keywords:** ethics of care; corporeality; the body; biopolitics; deconstruction; ethics of needs

## 1. Introduction

"The challenge is to think of 'need' in terms other than mere lack, as other than a barrier to well-being. Can we think of a need as something other than what must be overcome or satisfied prior to engaging in activities that provide real rewards?" [1] (p. 468)

"Care is omnipresent, even through the effects of its absence." [2] (p. 1)

At the time of writing this article, we have lived with the COVID-19 pandemic for over two years. In the privileged part of the world where I live, everyone over 12 years old who wants the vaccine has been vaccinated. Booster vaccinations have been delivered. With high vaccine rates turning the disease into something less incapacitating and deadly, the threat of health systems collapsing starts to pass. Societies are being re-opened, and traveling is again possible. But at the same time, these very same vaccines that protect our privileged bodies and societies remain inaccessible in large parts of the world. According to the World Health Organization, just under 71% of the population in high-income countries were fully vaccinated in mid-February 2022, while in lower-income countries, the same applied to only 6.6% of the population [3]. This is not because of any lack of capacity in the world to produce an adequate number of vaccines to get everybody protected. No—it is just world politics as usual, in an era of transnational capitalism.

Even though the only way out from the pandemic would be to vaccinate the entire global human population as a matter of urgency, many wealthy states have chosen a route of 'vaccine nationalism': i.e., hoarding vaccines for their own populations, opposing the waiving of patents from Pfizer, BioNTech and Moderna (the pharmaceutical companies that produce the vaccines) and refusing to push companies to share their technology and know-how in order to increase vaccine production. While these companies have benefited from public finances when developing their technologies, their monopoly and private profits are now protected by states, in the spirit of a (neo)liberal laissez-faire ideology.

Simultaneously, the three vaccine-producing companies have been seen to gain profits at the rate of 1000 US dollars per second, and in 2021, the companies made some $34 billion in profit between them [4].

For those familiar with the transnational distribution of care resources and the political economies involved, there is nothing new in this situation of global vaccination politics. In an era of transnational neoliberal capitalism, care provision and access to adequate care are increasingly dependent on the worth of the care-receiver to the accumulation of transnational capital. This means that those bodies and populations whose care needs are tied to financialized health care systems feeding into transnational capital accumulation are responded to with care. Simultaneously, the needs of those *whose care-receiving cannot be turned into profit* are increasingly neglected in the field of public care provision. In fact, neglect itself may be required for transnational capital accumulation to thrive. This is the case, for instance, when a pandemic is *not* stopped by providing vaccines to the entire global population, even though this would be possible. This protracts the pandemic, while at the same time creating a continuous need for new vaccines and boosters, which in turn increase the profits of the vaccine producing companies.

The same logics of care delivery apply across the world and in all societies, where the needs of those whose care adds to transnational capital accumulation are responded to with care, and the needs of the rest are likely to be neglected. In different public health systems, the accumulation of financial profit through care provision takes place through various mechanisms, for example through insurance, public-private partnerships, and private companies selling care services to public providers responsible for the care of the population [5–7]. Neoliberal states allow this to happen when governing care resources in ways that are in line with this neoliberal reasoning. Thus, even in the world's strongest welfare states today, tax-funded public care services are increasingly being commodified and marketized, which allows financialized care-selling corporations to make profits for transnational shareholders on the back of the citizens' care needs [5,6,8].

In terms of COVID vaccines, it would appear that the vaccines were developed quickly for this very same reason, and then distributed unevenly and selectively among the global population. The bodies that have now been protected with the vaccines are part of wealthy populations in states, whose public health financing benefits transnational capital in various ways, including globally privileged vaccine deals. In contrast, as Dr. Ayoade Alakija (Co-Chair of the African Union's Vaccine Delivery Alliance) put it in an interview with the BBC after the Omicron variant was first detected in South Africa: "Had the first SARS-CoV virus originated in Africa, the world would have locked us away and thrown away the key. There would have been no urgency to develop vaccines because we would have been expendable [9]". Her claim might not apply to all of the heterogeneous countries and populations in the African continent, yet it is likely to apply to those health systems that are unable to finance vaccine development and delivery in ways that benefit transnational capital.

As pointed out by several political theorists of care [5–7,10–12], such entanglements between care provision, care needs, and transnational capital accumulation are embedded in neoliberal discourses of governmentality that emphasize individual autonomy, independence, and rational economically driven decisions regarding the care of the self. In her recent work, Joan Tronto has criticized care theorists for treating this hegemony of neoliberal political ideology as invincible, and for not looking for alternatives. She proposes that instead of "trucking with *Homo oeconomicus*"—the economic man—as the main form of political subjectivity, we should envision "a different route to greater political concerns, one that begins with acknowledging that humans are essentially, in the plural, *Homines curans,* 'caring people'" [13] (p. 28).

I side with Tronto's call for alternatives. However, while I support and agree with her visions of *Homines curans* as providing a basis for the kind of political order that she calls a "caring democracy" [13,14], in this article, I seek to provide a theory that erodes the neoliberal ideological hegemony from a slightly different, although complementary angle.

Building on my previous deconstructive readings of feminist care ethics [6,15], I argue that care ethicists' traditional focus on practices of car*ing* as the origin of moral and political relations is inadequate in resisting neoliberal forms of governmentality.

Inspired by the work of moral psychologist Carol Gilligan [16], classic care ethics argues that caring practices provide a unique source of moral and political thinking, that recognizes human vulnerability and interdependency as existential and shared conditions of humanity. While challenging the (neo)liberal political ideal of individuated autonomous subjectivity, the emphasis in care ethics on practices of car*ing* as originary to political and moral relatedness makes it incapable of fully addressing the political power of bodies that depend on care. I have demonstrated this previously through a close reading and deconstruction of Joan Tronto's groundbreaking work in the field of political theory [17,18]. However, the same argument would apply to all care ethics that put car*ing*—rather than e.g., embodied practices and habits of needs—at the center of the theory. This could include, for example, the important work of scholars such as Selma Sevenhuijsen [19], Victoria Held [20], and Fiona Robinson [21,22] to name but a few.

In this article, I continue to argue that an ethic of care that is framed through car*ing* cannot fully address the power that bodies have—as bare, needy bodies—in shaping the world. This is because the kind of care ethics that emphasizes car*ing* ties political agency (even of *Homines curans* in caring democracies) to subjects that have the capacity to care. Thereby, the political power of "bare" bodies in need of care is written out and left unexamined, inarticulate, and mute. To tackle this problem, in this article I develop an *ethics of needs* that is complementary (i.e., not to be seen as a replacement) to the traditional accounts of care ethics. The concept of "bareness" in my ethics of needs comes from Giorgio Agamben's view of the modern politics of life (i.e., modern biopolitics), where life is divided between *bios,* the politically valuable life, and *zoē,* the "bare" life that lacks political value and can be killed without sacrifice [23]. In the neoliberal epistemic order, those lives in need of care, whose care cannot contribute to transnational capital accumulation seem to emerge as bare life that can be left to die without wider loss. To resist this particular violence imbued in the neoliberal ideology, I believe we need an ethics of needs, where the field of *zoē* emerges not as *bios,* but as something politically valuable in its own terms.

In my article *The Power of the Vulnerable Body: A New Political Understanding of Care,* I deconstructed Tronto's political care ethics through Giorgio Agamben's biopolitics in the context of the Finnish welfare state [15,23]. There, I analyzed the Finnish aging population as "bare life" that is capable of exerting "pressure on the sovereign power" in its bare neediness, while "forcing the state to adjust its political economy to the citizen's bodily needs of care". In this account, the "frail" and seemingly apolitical care-dependent bodies of persons with dementia emerged as politically powerful—even though they no longer have the capacity to practice car*ing*. Their "bare" yet globally *powerful needs* were capable of putting in motion entire political processes that could, by means of transnational biopolitics, reach across the globe [15] (p. 100). This happens, for instance, through the practices of global nurse recruitment, the political economies of which I analyzed in detail in *The Global Biopolitical Economy of Needs* [6].

These deconstructive readings of classic care ethics have given rise to my definition of *care understood as a corporeal-relation-enacted-by-needs* (see also [11]). This conceptualization of care decenters the focus of care ethics from practices of car*ing* and care labor while bringing the power of the needy body to the center of political life. Effectively, in deconstructing care ethics at its own limits, the concept of care as a corporeal-relation-enacted-by-needs opens a space for an ethics and politics where the body in need of care gains a central position, while at the same time decentering the subjective, caring "I".

In this article, I continue this previous work and start to develop a new care-ethical thinking, namely, an ethics of needs. It is, however, important to note I am deeply embedded in the classic tradition of care ethics, too. My aim is not to replace it with a better theory, but to extend and complement it through deconstructive exercises. The aim of deconstructive reading is never to destroy or reject the tradition one writes about. Rather,

it is about undertaking an intimate reading of the tradition from within, with the aim of working with its margins, exclusions, and intimate others, and with the ethical motive of rewriting the tradition into something more inclusive of its previous Others [24,25]. As such, deconstructive reading is never seen as complete, but as an opening that calls for further deconstructions. Thus, while in this article I seek to rewrite the ethics of care into an ethics of needs, the new ethics of care needs that I propose is but an *aporia* or a call for further deconstructions of care from the perspective of bodies in need of care.

The ethics of needs that I propose emerges along with my theorization and writing. Yet, some (arguably limited and deficient) elaborations and delineations are required vis-à-vis previous literature. To begin with, the ethics of needs that I propose in this article is not the first theory for an ethics of needs in the field of care theory. Sarah Clark Miller's book *The Ethics of Need: Agency, Dignity, and Obligation* (2012) is an extensive philosophical exploration of the nature of needs, about "which needs are morally salient", and "what moral agents ought to do when they encounter others with such needs" [26] (p. 1). To determine our moral obligations to meet the needs of others, she combines Kantian ethics with feminist care ethics. In so doing, her account, too, thus eventually focuses on the moral-political power of car*ing*, whereas I seek to focus on the power of embodied needs. Furthermore, I draw on theories that could be seen as being critical of Kantian positions, and our philosophical premises are therefore somewhat different and possibly in tension with one another. Due to these differences, and since my account of the ethics of needs has emerged independently from Miller's work, in this article I do not engage with this previous ethics of needs, as much as I would love to. I do believe such engagements would be useful and necessary. However, I must reserve the space of this article to develop and articulate my own theory for an ethics of needs, however lacking it may be in its engagement with previous discussions on the politics and ethics of care needs. Thus, the time and space needed to engage my ethics of needs with that of Sarah Clark Miller, or to discuss the relationships of these with, say, Nancy Fraser's "politics of needs interpretation" [27] will hopefully come later, in some other space of writing and thinking.

The ethics of needs that I propose in this article is radically corporeal. As such, it complements the work of those care theorists and care ethicists who explicitly focus on the body. Maurice Hamington, for instance, has argued that "care cannot be fully understood without attending to its embodied dimension", and that "care can be a viable component of social ethics only if its corporeal aspect is addressed" [28] (p. 4). For Hamington, care is "corporeal potential realized through habits" [28] (p. 5), and it "flows from the knowledge manifested in the body" [28] (p. 39). I believe that such embodied understandings of care are entirely compatible with the ethics of needs that I propose in this article.

Other care theorists who focus on the body have emphasized the material dimensions of care as body-work that necessitates physical proximity, co-presence, and corporeal encounters [29,30]. My ethics of needs is deeply influenced by these accounts, yet emphasizes embodied needs over embodied work and care encounters. Furthermore, building on biopolitical theories of governmentality, my ethics of needs draws corporeal connections between micro-level care encounters and macro-structures of transnational capitalism. Similar to Fiona Robinson's critical global ethics of care [21], the ethics of needs thus emphasizes that the corporeal relation of care is never just a linear, singular trajectory that connects the caregiver with the care-receiver, but rather a vehicle through which needs shape political economies, both locally and across societies [6,11,15,31].

Importantly, however, my ethics of needs also emphasizes that care, when seen as a corporeal-relation-enacted-by-needs, always entails the potential of neglect or other forms of violence. Siding with recent accounts in care theory and critical disability studies, the ethics of needs recognizes that care can be non-innocent [32] and destructive [33], may include violence [34,35], or at least is "not necessarily diametrically opposed" to it [36]. In this regard, the ethics of needs goes beyond care and does not necessarily reach the event where car*ing* takes place. Thus, in contrast to the ethics of care that tends to begin its analysis of political relatedness at the point where caring happens, the ethics of needs

begins where bare needs exist—also when they fail to enact relations of care. Following Erin Manning's theory of relatedness, the ethics of needs thus understands the relations that our needy bodies enact as "reaching-toward" intervals that are "active with tendencies of interaction", while never limited to interaction. They are "quantum leaps occur[ing] in a fractal mode of relation where events build on events" influencing simultaneously both the body in need and its environment" [37] (p. 34).

Philosophically, the ethics of needs that I propose builds on Jacques Derrida's philosophy of deconstruction, as well as Gayatri Chakravorty Spivak's notions of subalternity and epistemic violence. Much like critical disability scholars such as Margrit Shildrick [38,39] and Robert McRuer [40], I argue that in the present era of transnational capitalism, the body in need of care is the subaltern of neoliberal biopolitics and remains partially so even in the traditional accounts of the ethics of care. To provide an alternative account that complements this lack in care ethics, I embark on the task of writing the "subject under erasure" in relations of care. This, I argue, is necessary for us to envision how our embodied care needs and their power differentials silently write the world while subverting both the rational *Homo oeconomicus* as well as the subjective, caring "I".

The ethics of needs is therefore an ethical urge to try and read the needy bodies' subaltern writing of the world, even when its messages remain inarticulate and incomprehensible in the prevailing epistemic orders and the care relations inscribed in (and by) these orders. In developing the ethics of needs, my ambition is to provide an additional alternative to the hegemonic ideology of neoliberalism: a new grammar by which the neoliberal readings of the world could be eroded—also at times and places where care relations do not emerge when they should or could emerge. The erosive work of *Homo egēnus*, the needy human being, therefore complements the work of *Homines curans*, in striving for a political order where caring *and* care needs are central elements of emergent political orders.

My theorization in this article proceeds as follows. In the section following this lengthy introduction, I discuss Derrida's philosophy of deconstruction and *différance*, in order to provide the conceptual tools needed for reading my theory of an ethics of needs (which I admittedly still struggle to articulate comprehensively and in an elaborate fashion). Here, I also sum up Spivak's theory of subalternity and epistemic violence, drawing on Spivak's engagements with Derridean philosophy. In the third section, I return briefly to the Foucauldian vision of neoliberal biopolitics, so as to describe how life (including care and care needs) is governed in the present neoliberal episteme of transnational capitalism. Siding with critical care ethicists, I argue that a most pressing moral problem with neoliberalism is that it cannot caringly accommodate death in its politics, nor other forms of life that are dying and fully dependent on care for survival. This renders bodies-in-need as an excess to politics and political life in neoliberal capitalism, or as "bare life" in terms of Agambenian biopolitics [15,23].

The body in need of care thus emerges as the subaltern of our present epistemic order: it is that which cannot be heard and is muted, partially so even in care ethics. In order to read the writing through which needy bodies write the world, a new ethics of needs must be articulated. To do so, and following the Derridean philosophy of deconstruction, I write the subject under erasure in the fourth section of the article. Here, I articulate my theory of the lowest common denominator of embodiment, which demarcates the level of embodiment where the power differentials between bodies' needs can be traced and analyzed. As elaborated in the fifth and concluding section of the article, the body-organism-in-need-of-care now emerges as *différance,* and as a bundle of relatedness that constantly (re)writes the world with the corporeal relations of care *and* neglect it enacts. To conclude, I articulate the ethics of needs as an ethical position that seeks to read the world that embodied care *needs* write, with a particular focus on those needs that are presently responded to with neglect, indifference, or even violence, rather than care. Specifically, the ethics of needs opens a space or *aporia* for a more ethical politics of life than neoliberal biopolitics can ever provide, namely, the politics of *life of needs*.

## 2. Conceptual Tools

### 2.1. Deconstruction, General Writing and Différance

To understand how the ethics of needs demands an ongoing deconstruction of the world through silenced care needs, we need to return to the philosophy of deconstruction, as devised by Jacques Derrida. In short, deconstruction denotes the practice of a systematic tracing of silences and suppressed meanings in texts. It involves a double move, where one first works with the dichotomies on which meaningfulness relies to undo and displace their hierarchical opposition so that in the second move of deconstruction, the terms of these hierarchical binaries can eventually be situated anew [24,25,41,42].

In the philosophy of deconstruction, the term "text" does not refer to writing in the traditional sense of the term. For Derrida, "there is nothing outside the text" [41], and following Saussure [43], Derrida understands language as a system of differences. Words have a meaning only in relation to other words in a system that is defined in relation to other signs. Unlike Saussure who considered the spoken word as the pure form of language, for Derrida, this applies not only to spoken language. In fact, Derrida criticizes Saussure for prioritizing speech over writing. He shows how by suppressing the written language, Saussure prioritizes the Western metaphysics of presence over absence: disregarding that which is written out from language for the language to make sense in the first place. As a system of differentiation, language can only make sense in relation to the absented [41] (p. 46). In deconstructing Saussure's work (and thereby, the entire Western metaphysics of presence), Derrida produces an understanding of generalized or *arche*-writing. Here, generalized writing does not just take over the primary position that was previously held by spoken language, but rather spoken language already belongs to what Derrida calls generalized writing [41] (p. 55).

Due to its emphasis on "texts" and "writing", Derrida's deconstructive philosophy is all too often (mis) understood as dealing merely with language and textuality on an "immaterial" level of discourse. However, to understand how the care needs of bodies write the world, it is pivotal to understand that the generalized writing does not refer to writing in its literary, generic sense [44–46]. As Vicki Kirby elaborates, with reference to Derrida [41] (p. 9), when writing is understood in its most general sense, even "'the most elementary processes of the living cell' are a 'writing' and one whose 'system' is never closed" [45] (p. 61). Thus, Derrida's "notion of writing cannot be understood literally", and this is because "it makes the world, objects and relations possible; it structures and gives the world and its contents meaning and value" [44] (p. 84). Such writing of the world (s) "is itself a silent play" [47] (p. 5).

The ethics of needs is therefore an ethics attuned to reading the writing or 'play' of the most silent and silenced care needs, with an effort to decipher *how these needs and their power differentials write* the world we live in. To comprehend this playfulness of generalized writing, we need to revisit the meanings of *différance* in Derrida's philosophy. *Différance* is a neologism coined by Derrida, which denotes the vehicle of temporized movement of differentiation in texts. It is "literally neither a word nor a concept" [47] p. 3, and the precise definition of *différance* is therefore impossible. "[T]here is nowhere to begin to trace the sheaf or the graphics of *différance*", Derrida writes, "[f]or what is put into question is precisely the quest for a rightful beginning, an absolute point of departure, a principal responsibility" [47] (p. 6). Nevertheless, to elaborate on the play of *différance,* one must begin somewhere.

A semantic play on the French verb *différer* forms the (impossible) origin for the (non)concept of *différance*. While in English, the Latin verb *differer* translates into two separate words—to defer and to differ—in Derrida's mother tongue, French, the single verb *différer* holds both these meanings. Here, the first refers to deferring, i.e., "the action of putting off until later, of taking into account, an operation that implies an economical calculation, a detour, a delay, a relay, a reserve: temporization" [47] (p. 8). The second meaning of *différer* is then the "more common and identifiable one: to be not identical, to be other, discernible etc."—i.e., to differ. Thus, the word *différence* with an e cannot

simultaneously refer to both deferral and difference. As an economic compensation for "this loss of meaning", Derrida invented the neologism *différance* (with an a), which, he maintains, "can refer simultaneously to the entire configuration of its meanings" [47] (p. 8). *Différance* can do so precisely because it is not a concept or a word, but a "sheaf": a bundle, which has a "complex structure of a weaving, an interlacing which permits the different threads and different lines of meaning—or of force—to go off again in different directions, just as it is always ready to tie itself up with others" [47] (p. 3).

Influenced by the writing of feminist disability scholar Margrit Shildrick [38], I suggest in this article that the body in need of care is *différance,* operating in exactly these ways when enacting corporeal relations (of care and neglect) in the world. Through the relations it enacts with others, the needy body is always a "sheaf" and a "complex structure of weaving" that is "ready to tie itself up" with other corporeal beings [47] (p. 3). As I shall explain, this is so also when the response-able respond with neglect, indifference, or even violence, and when the epistemic order makes it impossible to recognize the needy as needy, or worthy of caring responses. To understand these relations of neglect—care in its absence—and the silencing of needy bodies in neoliberal biopolitics as well as in the ethics of care, it is necessary to return to Spivak's notions of the subaltern and epistemic violence.

### 2.2. The Subaltern and Epistemic Violence

As emphasized by Derrida in the entirety of his corpus, and Spivak in her introductory essay to *Of Grammatology,* the sheaf of *différance* takes as many forms as there are texts [48] (p. xv). This means that in all texts there is a silenced, heterogeneous Other, whom deconstruction aims to reveal as an erased trace in the text, with attempts to rewrite the text into a form that makes space for the Other in its complete Otherness. This haunting, speechless Other is *the subaltern*. It is marginalized and written out, and yet crucially central to the text since, without its very erasure, the epistemic order of the text would fall apart, and the text would no longer make sense. Thus, the erasure of the subaltern gives epistemic orders their sense and meaning.

In her widely cited essay "Can the subaltern speak?", Spivak argues that the subaltern subjectivity is utterly incomprehensible, irrational, and mute [49]. As a constitutive outside of the episteme, the subaltern cannot speak without simultaneously being assimilated into the order and thereby losing their own history and being. This is because to be comprehensible, the subaltern ought to speak the language of the hegemonic episteme—a language that remains intact only insofar as the world of the subaltern Other is muted. In this order, the voice of the subaltern does not make sense: it is irrational. The epistemic order is, by definition, an order that is founded on constant *epistemic violence* against the subaltern Other, who must forcefully and repeatedly be written out from the order, so as for the order to make sense in its constitutive rationality.

The trace of the absented subaltern is, therefore, *différance* written throughout the sign system of the episteme, as that which temporally defers and spatially differs from what makes sense. From the margins of the order, it continues to haunt the order's completeness. This play of the subaltern as *différance* has compelling consequences for the ethical reading of the world-as-text(s) [24,50,51]. As a trace of the haunting absence of the subaltern that differs in meaning and defers presence, for the meaningful presence of something/someone/somebody else to be possible, it keeps the material-discursive world open for subversion. *Différance,* in other words, renders all writings of the world incomplete and open, making it possible and obligatory to read the world otherwise or in a different way. Especially, it underlines the necessity to deconstruct all texts and all epistemic orders, including the ethics of care.

In Spivak's reading of Derrida, it is revealed how the philosophy of deconstruction is an ethical response to epistemic violence. Here, deconstruction begins with the recognition that all texts are based on an epistemic violence that erases the Other, while rendering the subaltern mute. Margins are never simply margins, but simultaneously "the silent, silenced center" [49] to which deconstruction recurrently returns. Deconstruction is about the



constant imperative to recognize the muteness of the subaltern as "that interior voice that is the voice of the other in us" [49] (p. 294). Deconstruction means constantly working with epistemic violence, while simultaneously "presupposing [a] text-inscribed blankness" [49] (pp. 293–294) that is the silenced subaltern.

However, this does not mean that deconstruction can remove all epistemic violence to make space for the subaltern to speak. The deconstructed new epistemic orders will also only make sense by writing something (and consequently some bodies) out, by erasing the constitutive outside that is its subaltern Other. Here again, the subaltern remains mute and continues to haunt the deconstructed realities. In an interview with Jean-Luc Nancy, Derrida himself underlined that the "deconstructive gesture" is always summoned by a "surplus of responsibility", which "is excessive or it is not a responsibility" [52] (p. 286). Here, justice is never completely gained, which means that the responsibility to try and hear the Other remains, even when they cannot speak. In the following section, I turn to elaborating how the body in need of care in neoliberal biopolitics is the subaltern that cannot speak—and whose writing of the world we are nevertheless excessively responsible to try and read.

### 3. The Body in Need as the Subaltern of the Neoliberal Episteme

#### 3.1. Neoliberal Biopolitics and Care

In his 1978–1979 lectures on the birth of biopolitics, Michel Foucault critically describes the operation of neoliberal governmentality as a means of governing populations through the embodied lives of individuals, and their rational choices [53]. Governmentality for Foucault is the "art of government", which directs the subjects to lead their lives in certain ways rather than others. The technologies of governmentality involve not only legislative and disciplinary measures but also discursive tools that govern "the conduct of conduct" among the population, in the most mundane, embodied level of their everyday practices of living.

The Foucauldian account of neoliberal biopolitics provides a useful prism through which to criticize the politics of care in neoliberal societies in an era of transnational capitalism [10,11]. Emphasizing the governance of populations through the practices of self-care of autonomous individuals, Foucauldian biopolitics reveals how in its recognized spheres of governmentality, neoliberalism mutes the recurrent moments of life where subjects are care-dependent and needy of others. This makes the neoliberal worldview illusionary and unrealistic [10–13]. Care ethicists, for instance, have emphasized that human beings are vulnerable and factually dependent on receiving care from each other [1,17,18,21]. Embodied care is a species activity without which humankind would perish [18]. While this should be self-evident, unfortunately the neoliberal politics of social and health care are increasingly being framed as if this were not the case, and as if the subjects and objects of care were disembodied, independent, self-caring actors making rational decisions throughout their autonomous lives, irrespective of their decaying bodies that start to age from the moment they are born [10,11].

The critical disability theorist Robert McRuer has argued that global neoliberal capitalism relies not only on an illusion of disembodied political subjectivity but simultaneously on the "compulsory nature of ablebodiedness". He explains that our freedom to sell our labor and flourish through economic productivity effectively means that we are "free to have an able body but not particularly free to have anything else". However illusory it may be in our factually needy material existence, this compulsory ablebodiedness is deeply interwoven with neoliberal conceptions of how to be human, and consequently with the political orders that emerge with those conceptions. "[W]ith the appearance of choice", writes McRuer, compulsory ablebodiedness covers over "a system in which there actually is no choice" [40] (p. 8).

However, as Tronto's notion of *Homines curans* emphasizes, in political life there always is a choice and an alternative [13]. Together with Hanna-Kaisa Hoppania we have argued that the very paradox within neoliberal governmentality—between the real-life embodied

demands of care and the illusionary neoliberal rationality by which care is governed—reveals the fragility of neoliberalism. Producing endless frictions in the neoliberal political economy, the embodied demands of care and the relations they enact keep the order open for the political, open for change, and open for alternative orders [11]. In this article, I wish to emphasize that if we take seriously both the ethics of care and ethics of needs and understand that *Homines curans* are also *Homines egeni* (needy human beings)—the change can be something radically different from the neoliberal episteme. However, to deconstruct neoliberal biopolitics, we need to first start re-reading it from within its own limits.

### 3.2. Dying Life as the Limit of Neoliberal Biopolitical Governmentality

In his last lecture of the 1975–76 lecture series *Society Must be Defended,* Foucault defines biopower as "the right to make live and let die" [54] (p. 241). In the same context, he indirectly comes to elaborate on why neoliberal biopolitics cannot accommodate life that is dependent on care from others for its own survival. He does this by addressing the question of aging and death as limits to biopower and neoliberal governmentality. These limits illuminate how the body-in-need is the subaltern of the neoliberal epistemic order.

Foucault argues that when calculating and preparing for moments where an individual becomes "incapacitated", put "out of circuit" and "neutralized", the neoliberal governance of life necessarily intervenes in certain universal and irreducible conditions of our embodied vulnerability. These risks, he argues, are particularly "the problem of old age, of individuals who, because of their age, fall out of the field of capacity, of activity" [54] (p. 244). In the field of care, we know that such "incapacitation" can happen at any time for various other reasons such as disablement, illness, or injury, and not just due to old age. Some individuals may never in their lifetime fall *in* to "the field of capacity", as defined by neoliberal reason.

According to Foucault, the neoliberal solution to this universally ever-present possibility of the body's "incapacitation" is the securitization of life through insurance and collective savings [54] (p. 244), which ensures the means to access care when the person is no longer capable of autonomous self-care. This is the logic by which present-day neoliberal societies operate in their governance of care. In her diagnostics of the neoliberal ideology in the politics of care, Joan Tronto names the techniques by which care needs are met in neoliberal societies: namely through responsibilizing individuals to take care of their own needs, by turning to markets for care provisioning, or by relying on family members to provide for their needs [13] (pp. 29–30). However, none of these technologies of governing life can fully tame the biopolitical problems of aging, dying, and "incapacitation". Ultimately, even if a body is insured for the costs of care, capable of purchasing care from the market, or able to rely on familial relations of care, eventually all life must give way to death. And death and the processes of dying—the moments where self-caring capacities and the productive potential of life escape governmentality—are something that neoliberal biopolitics cannot fully capture. Death becomes "the most private and shameful thing of all", the ultimate taboo "to be hidden away". In neoliberal biopolitics, death emerges as "the moment when the individual escapes all power, falls back on himself and retreats, so to speak, into his own privacy. Power no longer recognizes death. Power literally ignores death" [54] (pp. 247–248).

In erasing death and dying life from within its governmental reason, the neoliberal politics of life (i.e., biopolitics) is incapable of addressing those bodies that would die without care, and that may no longer be made to live the kind of life that is productive, and therefore, a life that is worthy of living according to the neoliberal rationality. Neoliberal biopolitics, I argue, relegates these lives to the apolitical field of *zoē*, as bare life, whose life and death are a matter of indifference: they can be killed—or, as Foucault might put it, be let to die—without sacrifice to the capitalist economy, which is the *sine qua non* of neoliberal governmental reason [6,15,23]. Consequently, in the epistemic orders of neoliberalism, the bodies in need of care—i.e., those who can no longer be turned into rational, choice-making, self-caring subjects—cannot speak.

Thereby, the dependent body-in-need emerges as the subaltern of the neoliberal episteme: Whenever a person is fully dependent on care for survival, without the promise to re-establish themselves as a self-caring subject, they become potentially meaningless for today's hegemonic epistemic order. This is particularly so at times when the body's care needs are no longer capitalizable and economically productive [55], for instance, when care-provisioning does not add to transnational capital accumulation through insurance or commodified care markets. Thus, in the present neoliberal epistemic orders, embodied care needs often operate as an erased yet central trace, or as Spivak might put it, as "an always already absent present, of the lack at the origin that is the condition of thought and experience" [49].

That care-dependent bodies are muted by default enables injustices in the field of care, which cannot be tackled because they are non-representable. Therefore, the subalternity of the body in need of care is a central field of epistemic violence in our neoliberal times. This concerns us all. Namely, when it is recognized that all human beings live in body organisms that cannot survive without care from others, the lives of us all are partially written out from the neoliberal political order, even if differentially so. In the following section, I elaborate on how care ethics also partakes in this epistemic violence by partially muting the subaltern body-in-need.

### 3.3. The Muted Body in Care Ethics

Rosi Braidotti has described the Foucauldian biopolitics' incapacity to address death and dying life as a "residual type of Kantianism" that emphasizes the individual's responsibility for the self-management of one's health, care, and life in general. Braidotti goes on to argue that the downside of this position is that such a politics of life "perverts the notion of responsibility towards individualism, in a political context of neo-liberal dismantling of the welfare state and increasing privatization" [56]. Care ethics provides a realistic challenge to this individualism, and thereby to the neoliberal rationalities by which societies (and the care relations therein) are presently being governed. Showing how morality and politics do not ensue from rational decisions over self-care made by illusionary, disembodied, autonomous individuals, but from situated and embodied practices of care in response to the needs of others, care ethics uncovers the paradoxes and limits of neoliberalism. Thereby, care ethicists continue to challenge the very foundations of (neo)liberal thinking.

There is, however, still a particular liberal "residue" in care ethics. Namely, as discussed in the introduction, by emphasizing moral and political relations as emanating from the practices of car*ing*, care ethics foregrounds the political value of those embodied actors in care relations who have the individualized capacity to care and respond caringly to the needs of others. Chris Beasley and Carol Bacchi have argued that this "paternalist protectionism that care ethics invokes represents only a partial challenge to neoliberal thinking" [57] (p. 285). To me, this is symptomatic of the ways in which many classic accounts of care ethics also mute the needy body. To further examine this argument, let us briefly return to Spivak's theorization of subalternity.

In elaborating on the muteness of the subaltern, Spivak has argued that the hegemonic episteme operates "its silent programming function" through all segments of the population [49] (pp. 282–283). This means that all subjects of the epistemic order partake in muting the subaltern, and insofar as one seeks to make sense within the grammars of the order, it is practically impossible not to. In her essay "Can the subaltern speak?", Spivak showed how even the allegedly most radical critics of the prevailing episteme can end up silencing the subaltern. Spivak's criticism in the essay was addressed to two celebrated leftist intellectuals, namely Foucault and Deleuze, and through their example, more generally to the Western post-structuralist theory. In her essay, Spivak deconstructs the seeming transparency of these intellectuals' positions in their critiques of the capitalist world order. She argues that even as Foucault and Deleuze in their critical analyses of the modern society seek to appreciate "the discourse of the society's Other", they still tend to present the Other as a monolithic mass—for instance, as "the Maoists" and "the workers".

Thereby, while the intellectuals were capable of naming and differentiating between other (Western) intellectuals, they simultaneously failed to recognize the heterogeneity within the otherness they claimed to know [49] (p. 272).

The danger (and epistemic violence) here is "the first-world intellectual masquerading as the absent nonrepresenter who lets the oppressed speak for themselves" [49] (p. 292), while the subaltern Other is simultaneously excluded from the epistemic order within which the critique is being articulated. Spivak emphasizes that whereas the transparent intellectual may well be in a position to make visible the mechanisms of oppression, rendering the oppressed 'speakable' within the prevailing discourse is a whole other question, and impossible in the case of the subaltern.

It is important to bear in mind the decolonizing aims of Spivak's argument, which I do not wish to co-opt in this article when utilizing her argument to underline the muting of the needy body in the neoliberal episteme. Spivak's discussion of the subaltern unmasked the role that the transparent Western intellectual has had (and often continues to have) in upholding the postcolonial economic order and its international division of labor. Spivak drew particular attention to the constitution of the female colonial subject as the Other in the specific context of India, asking whether and how she could speak. These aims in Spivak's theory of the subaltern are also crucial to the ethics of needs, even if the scope of this article is inadequate to fully account for the postcolonial and economic dimensions of this ethic. Consequently, in several ways (and not least due to the citational politics of this article), the ethics of needs as articulated in this article remains open for decolonial critique.

For the moment, however, I wish to utilize Spivak's critique of the Western intellectual to elaborate on the relationship between care ethicists as intellectuals, and the position of bodies-in-need within the ethics of care. As a care ethicist, I am also the object of my own critique. Indeed, my very need to try and articulate an ethics of needs as a supplement to the ethics of care is triggered by the necessity to address the needs of those bodies that my care ethics will inescapably always exclude, that is, the bodies that will forever haunt the ethics of care. The question here is: When the ethics of care sees moral and political relations as enacted by the practices of car*ing* (i.e., by the practices of subjects with an adequate psycho-social capacity to care), *can the bare-bodies-in-need speak in this ethics*?

Care ethicists recognize care recipients and their responses as central to the "circles of care" [18], and caring practices in care ethics are always responses to the needs of particular others. This gives care ethics its distinctive situated and political character, different from moral theories of abstracted justice. It would therefore be misleading to argue that care-recipients are entirely absent in the ethics of care. They are there, possibly in the margins as far as political agency is concerned, yet present as objects of caring practices as well as through their subjective demands and responses to care. However, when caring is a response to the needs of particular others, only those bodies-in-need whose *subjectivity as a care recipient* is recognized become included in the ethics of care. Those bodies that need care, yet lack recognized subjectivity altogether, or subjectivity *as needy* in established epistemic orders of care provision (often including the embodied needs of the carers), become written out as a mass of Others. The needs of such bodies cannot quite speak, and in the ethics of care, *needs* as *bare needs* cannot speak—only the *subjects* of caring and care-receiving can.

Thus, as care ethicists and intellectuals, we may recognize and see bodies-in-need as a corporeal, fleshy mass, muted in the circles of care. Yet, we still fail to decipher the heterogeneity of this mass and the power relations and differentiations therein. Thereby, in its quintessentially ethical demand to respond to the needs of particular, individuated, personified others, the ethics of care remains tied with the liberal emphasis on *individuated subjectivity* when the question of needs is considered. To the extent that the body-in-need (as a "bare" body stripped from its subjectivity) is the subaltern of neoliberalism that cannot speak, the ethics of care operates within the same epistemic order. Consequently (like Foucault and Deleuze in Spivak's critique of the Western intellectual), as care ethicists and intellectuals, we are unable to fully escape the rationality of the epistemic order we seek to

criticize. Instead, we remain partially and unintentionally "complicit" in upholding it, even in the most radical criticisms that emphasize caring as a challenge to *Homo oeconomicus* [13]. Therefore, as care ethicists, we need a complementary ethics: an ethics of needs that begins its analysis of relatedness from bare bodies deprived of their subjectivity.

Spivak writes: "In the face of the possibility that the intellectual is complicit in the persistent constitution of Other as the Self's shadow, a possibility of political practice for the intellectual would be to put the economic 'under erasure,' to see the economic factor as irreducible as it reinscribes the social text, even as it is erased, however imperfectly, when it claims to be the final determinant or the transcendental signified." [49] (p. 280) As care ethicists, we are inescapably complicit in the persistent constitution of the needy body as the Other in the shadow of our caring Selves. This inescapable othering is written in the grammar of the ethics of care, where it cannot be avoided. The subaltern Other here is not the *subject*-in-need, but the (bare) body-in-need, deprived of its individually articulated subjectivity. Therefore, the subject (and not simply the self-caring individual as in care ethicists' criticisms of neoliberalism) is the economic reason for our epistemic order that must be written under erasure. This I start to do in the next section.

## 4. Analyzing Relatedness on the Level of "Bare" Needs

### 4.1. The Lowest Common Denominator of Embodiment

In the ethics of needs, each embodied trajectory of human life is recognized as unique, defined by particular and innumerable corporeal relations of care and neglect, which are enacted by needs and that extend through both space and time. From one embodied life to another, the quality and quantity of care, as well as neglect, differs. This is because the care needs of our bodies are always differentially powerful when compared to all other bodies at any moment of time. The need is furthermore never entirely powerless. It always enacts a response, although at times the response is not a caring one. The care ethicist Fiona Robinson has argued, that "relations of care are constructed by relations of power determined by gender, class, and race. These are, in turn, structured by the discourses and materiality of neoliberal globalization and historical and contemporary relations of colonialism and neo-colonialism. In this view, thinking about care in the context of global politics and security cannot posit a universal need for care as unproblematic and undifferentiated; *needs are themselves constructed and produced by a wide range of relationships and structures.*" [21] (p. 5, emphasis added).

Here, the ethics of needs adds that care needs are not only *produced by* various relationships and structures. Embodied care needs and the power differentials between them are also *productive of* the governmental relationships and structures, within which they demand a response. These power differentials are not only inscribed in our intersectionally gendered bodies, but our needy bodies also inscribe the world. They do so by the relations that their differentially powerful needs enact with other bodies. Yet, it is only after recognizing all bodies as existentially needy of care from one another, and all needs of care as capable of enacting political relatedness, that analyzing the differential power of needs becomes possible at all. When it comes to the body's ontological need for care, there is no exception, only difference.

But how does one read difference and the power relations between "bare" bodies-in-need, that lack the subjectivity of a speaking, caring "I"? The ethics of needs focuses on needy bodies at the level of what I call *the lowest common denominator of embodiment*. Hamington has pointed to the body as a common denominator of humanity [28] (p. 39), but in the ethics of needs, I wish to be somewhat more specific. With the lowest common denominator of embodiment, I refer to the fact that as living and dying organisms, all human bodies have certain mundane material needs that make humans existentially dependent on one another. The lowest common denominator of embodiment is the physiological fact that, in addition to breathing, all human bodies need to be fed and watered, and need to digest, as well as discharge urine and excrement in an adequately hygienic way. These are daily needs, and when they are not met, the human being in question will eventually die.

While there may also be other fundamental needs, I refer to these needs because they apply to every single living/dying human body, every day, from the moment of birth to death, regardless of age, gender, sexuality, race, ethnicity, class, status, or any other attributes of identity, or social position. Indeed, if we just focus on the fact that every human body must eat, drink, urinate, defecate, stay clean, and require care from others when not capable of doing so independently, the lowest common denominator of embodiment marks a space where differentiating the feminine from the masculine is not self-evident. Hence, as living organisms on the level of the lowest common denominator, our bodies are needy as well as fundamentally queer.

From this perspective, no human being can exist without care provided by other bodies: and in order to survive, we all must eat, drink, poo and wee, and stay clean, and over a lifetime this requires care from other bodies if we are to stay alive. Just as in the ethics of care, in the ethics of needs, care is still a species activity [18], and we are never truly individual as our inescapable neediness makes us always already related to and with other bodies. With this argument, the ethics of needs seems no different from the ethics of care. And yet, it is. Namely, in the ethics of needs, also other responses to needs than the caring ones count: Neglect is a species activity, too.

The lowest common denominator of embodiment refers to the analytical level of embodied human life where we are stripped of articulate subjectivity—i.e., our capacity to speak, either literally, or in ways that make sense in the prevailing epistemic order. It is a level of analysis where we are but animals, with the socio-political relationality of humankind, and its structures of governance emerging with the relationships that our barest needs enact with other bodies in the world. Such relations emerge because bodies can be (and very often are) both needy and caring. That our lives are defined by such a dimension of bare-neediness on the level of the lowest common denominator does not mean that other levels and dimensions of life would simply go away. As Hamington puts it: "there is no essence of embodiment that can be separated entirely from the body's transactions with society" [28] (p. 41). Hence, on the level of the lowest common denominator, the subject positions and identities that are inscribed on bodies by the governmentalities of prevailing epistemic orders still matter. The power differentials between bodies' capacities to enact relations of care or neglect are shaped by such inscriptions.

For instance, if inscribed with the citizenship and rights of a welfare state, a person living with a severe form of dementia, having lost their capacity to articulate themselves as a speaking, thinking, or caring "I" in the prevailing episteme, may still rather powerfully enact care relations. They may put in motion political processes and legislative changes by demanding the publicly funded services that bodies in that society are eligible to access. Their bodies (even if in many ways muted) demand vaccines and boosters in the time of a pandemic, when entire populations elsewhere are left without. Such bodies can make care resources and caring bodies move, even across the globe, when migrant care workers are utilized to fill care deficits in aging welfare states [6,15]. These powerfully needy bodies are not only inscribed with rights and wealth, however. Their bodies, as bare, needy bodies, are also inscribed with *value* that can be extracted for transnational capital accumulation through care provision. This is because the neoliberal welfare state's financialized structures of care provision tie public spending on care needs with the interests of transnational shareholders of care-selling businesses. This may apply to situations where financialized care providers owned by private equity firms sell services to the public sector [5,8], as well as to global vaccine politics where vaccine nationalism feeds into the profits of pharmaceutical companies.

In short, bodies' care needs in this world are attended to when the needs are both economically and politically valuable. And yet, even here amongst an allegedly privileged mass of aging citizenry in wealthy welfare states, needy bodies are never just about a bare mass. There are power differentials between needy bodies, where inscriptions of racial, class, gender, and other differentiations come to matter, in just *how* powerfully the

bare-body-in-need (on the level of the lowest common denominator) can make the different networks, circles, and processes of care move, both globally and locally.

Simultaneously, a body-in-need with dementia also demands care in some less care-privileged part of the world (e.g., a homeless person in the US, or some other care-poor society where bodies are *not* inscribed with rights to care or publicly funded social security). This body, too, enacts relatedness by its very needs. The body may not have the power to instantiate relations of car*ing*, or when they do, the relations enacted are understood to be about charity rather than political rights. Yet, the body is still not powerless in its needs. Its needs, too, write the world with political relatedness. For instance, the body's needs feed into the structures of the transnational political economy. While responding to the body's needs *caringly* may not add to transnational capital accumulation, neglecting the needs may well do just that, when enforcing the idea that one should be financially insured against the need for care—or is not eligible to care at all. However violent and unjust it may sound, these relations matter, both politically and economically. Each relation enacted by needs is a unique opening and reaching out to the world that inscribes the world and its order at that particular moment of time. If we only focus on caring relations as is the case in traditional accounts of care ethics, we miss the opportunity to even try and read what other types of relations of needs do to and with the world.

I argue that as care ethicists, it is our moral responsibility to develop ways to read the relatedness that all kinds of needy bodies write, or else we remain partially complicit in reproducing the position of these bodies in neoliberal epistemes as a bare abandoned mass outside the established governmentalities of care. When focusing on human relatedness on the level of the lowest common denominator (bare speechless bodies related to one another through needs), the network of relatedness that emerges is quite different from the one that emerges with relations of car*ing*. To envision this network of needs, we need to move away from caring and write the subject temporarily under erasure.

### 4.2. Writing the Subject under Erasure

The ethics of needs focuses on bare-bodies-in-need-of-care. Such a focus necessarily obscures the psyche and lived experiences of those subjective persons who need care. In various seminars and conferences, I have been interrogated about this ethical dilemma: Is it not *unethical* to talk about the bodies in need without addressing their subjective agency? Does that not lead to a position where the needy are treated as a faceless mass lacking agency altogether? These concerns are undoubtedly valid in an epistemic order where the body-organism "in itself" (without a subjective speaking, caring, thinking "I", or as bare life of *zoē* outside the politically valuable life of *bios*) is understood as lacking political agency. However, the ethics of needs attempts to go beyond such an epistemic order. It seeks to comprehend the power relations and politics of human life in the field of *zoē*, where we always already exist as meaningful, unique, and valuable body-organisms, both before and after articulating ourselves as a subjective, caring "I".

The trouble is that once we let the subject speak, addressing the moral and political relations between bare-bodies-in-need in the field of *zoē* becomes impossible. After all, once addressing the subject, we are back in the field of *bios*—the political life constituted by muting the subaltern life of bare-bodies-in-need. Thus, for an ethics of needs to emerge, the subject must be decentered. The point here is not to deny subjective agency from the care-dependent, or to say that their lived experiences do not count—they do! In the ethics of needs, disregarding subjectivity is merely an analytical tool that might (just might) allow us to read power differences between the relations that differential care needs enact in the world. This, in turn, might (just might) also allow us to analyze those moral and political relations that bare bodies enact when caring fails to take place where it should or could but does not. In other words, it might allow us to examine the shadows of care ethics, where the moral power of care is present only in its own absence.

However, writing the subject under erasure is difficult. The Cartesian mind/body split remains one of the most powerful binaries defining our political life. As a remnant of moder-

nity in supposedly postmodern times, our political thinking continues to be organized as if our subjective minds owned our bodies: the (feminized) body remains devalorized and muted so the (masculine) rationality of the subjective mind can be cherished and heard. To an extent, this remnant is visible in the ethics of care, too. Even if it does not emphasize the rational mind of the *thinking* "I", the focus on a car*ing* "I" still remains.

In reality, of course, as feminist theorists of the body have shown for decades, the mind—be it caring, embodied, thinking, or speaking—does not own the body. We may well speak of the body "to others as of a thing that belongs to us; but for us it is not entirely a thing; and it belongs to us a little less than we belong to it" [58] (pp. 398–399, cited in [45] (p. 65)). However much we may valorize the rational mind as the basis of political subjectivity, our bodies continue to both pre-live and outlive the speaking "I" that is commonly understood as subjectivity: first in infancy and later for example through dementia or other cognitive conditions, illnesses, or injuries, or simply through the decay of the aging and dying brain. When subjectivity is considered from the perspective of embodied care, as pointed out by Hamington, "subjectivity is physical" [28]. The human body organism is an agent of its own, and also at times when it lacks an articulate subjectivity: an "I" that can speak, think, articulate, take responsibility, and care.

The ethics of needs focuses on political relations that bodies-in-need-of-care enact in the world. This is an ethics that seeks to read the ways in which care needs, as "bare" needs, write the world, and how they do it even when they lack articulate subjectivity. Because the writing of needy bodies takes place through the corporeal relations that *differentially powerful* needs enact in the world, the ethics of needs requires tools to read the power differentials between bodies as "bare" bodies that lack the subjectivity of the speaking "I". For this purpose, the ethics of needs focuses on bodies-as-organisms-in-need-of-care at the level of the lowest common denominator. This demands me *not* to address the subject.

Let us now briefly return to Derrida's philosophy of deconstruction, which involves the technique of writing "under erasure"—i.e., a writing method where words are struck out. As elaborated by Spivak, ~~the trace of the other~~ in the text for Derrida is not simply the inarticulable. It is *différance*, "the mark of the absence of a presence, an always already absent present, of the lack of the origin that is the condition of thought and experience" [49] (pp. xvii–xviii). In the predominant epistemic orders, ~~the-body-in-need~~ is ~~the trace-of-the-other~~ that cannot speak and cannot be heard. The ethics of needs is therefore a deconstructive move, where the binary between the Subject and its ~~erased Other in the body-in-need~~ is subverted by putting the subject temporarily under erasure.

To make space for writing about and with the body as a body organism with nothing but an ~~erased trace of subjectivity,~~ ~~the subject~~ must be written under erasure in the ethics of needs. This does not make questions about ~~subjectivity~~ and ~~subjective agency~~ simply go away, and indeed, it is imperative to address the subjective agencies of care-dependent persons in the wider politics and ethics of care. As elaborated in the previous section, from within the margins of the ethics of needs, various inscriptions of a ~~subject position and identity~~ on the body-organism still shape the power that bodies-in-need have in enacting relatedness with the world. The erasure of ~~the subject~~ is therefore but a deconstructive move that the ethics of needs makes for analytical purposes. The ethics of needs fails to make it a double move, in that the binary hierarchy between ~~the subject~~ and the body-in-need would be "'reinstated' with a complete "reversal that gives it a different status and impact" [25] (p. 150). Nevertheless, to examine how the body as "bare", needy human life writes the world with the relations its *needs* enact—and to read the power differentials between *needs* rather than ~~subjects~~—in the ethics of needs we must refuse to talk about ~~the subject~~ or *its* embodiments, even if that means leaving the deconstruction of embodied subjectivity in suspension. Consequently, the new episteme that emerges when ~~the subject~~ is written under erasure is an order written by the relations that needy bodies enact in the world in their differential power to enact relations of care. I will elaborate on this *différantial* play in the following concluding section, where I try and articulate how the

ethics of needs challenges the prevailing neoliberal episteme, while still complementing the moral positions of the ethics of care.

## 5. Conclusions: Envisaging a World Written by Care Needs

Vicki Kirby has argued that as scenes of writing, bodies always operate as their "own historical and cultural context[s]" [45] (p. 62). They carry within themselves complex trajectories of the past, while simultaneously being capable of enacting new histories and alternative politics when relating to other bodies. Here, some bodies are marked with subjective affects and pasts that help the bodies to enact relations of care, rather than neglect. However, due to a range of historical reasons and governmental technologies, revolting affects [59] such as disgust, abjection, and indifference easily stick to the bodies of others, which then tend to enact an aversive corporeal relatedness rather than relations of care. In such an affective "cultural politics of emotion" [60], historicized postcolonial scripts of racial difference materialize on bodies and their interrelations, as do scripts of dis/ability, gender, sex, age, wealth, and various other entanglements of differentiation.

Thus, each human body in this world produces and carries with it an ever-changing bundle of corporeal relatedness, enacted by differentially powerful care needs, while mobilizing care resources, caring bodies, and political economies of care in some ways rather than others. Be it the body of a globally mobile care worker, that of an older citizen with rights to public care in a welfare state, or a person dying of COVID-19 because of wealthy states' refusals of patent waivers; each human body lives and moves as an intersection of a unique set of corporeal relatedness. This weave of relatedness that the body both carries and produces simply through the processes of living is defined by the needs of the body itself, by the needs of the multiple others who demand care elsewhere, by the governmentalities that organize care relations and resources, as well as by the global political economies where needs are defined as differentially valuable for capital accumulation.

As further elaborated below, the ethics of needs is a theory of care that can be extended to more-than-human worlds [2]. Yet, if we stay with the human population for now, the ethics of needs imagines it as being comprised of almost eight billion bundles of relatedness, also known as human bodies. Of these billions of bodies, each needs care, and the needs of every one enact political relatedness in the world. As the innumerable corporeal relations elicited by these billions of needy bodies intertwine, each relation in tension with one another, what emerges is an entirely new articulation of the world and its changing relations of power. Here, the "bare" human body that is existentially in need of care from other bodies is no longer about a faceless, apolitical mass residing in the field of *zoē*. Instead, the body-in-need emerges as *différance*, that is, a sheaf of relations and potential politics that "permits the different threads of force to go off again in different directions, just as it is always ready to tie itself up with others" [47] (p. 3).

When understood as *différance*, the needy body is not just "a sign, a function of discourse", as bodies are sometimes understood in the feminist theorization of body writing [61] (p. 63). As *différance,* the needy body is a vehicle of temporized movement of differentiation. This means that the differentiation between bodies worthy of care and those that deserve mere neglect is never inscribed on the body from outside, in an overarching manner. Rather, as a vehicle of temporized movement of differentiation, the body in need of corporeal others constantly calls for the recognition of its needs. Importantly: *the differentiation takes place not on the body, or in the body, but in the emergent corporeal relations that the body in need enacts*.

Derrida argued that "*différance* is also the element of the same (to be distinguished from the identical) in which oppositions are announced" [62] (p. 9). This supports my understanding that power relations between bare-bodies-in-need should be examined at the level of the lowest common denominator, where all bodies need to stay adequately clean, eat, drink, urinate and defecate, and need care to do all that, albeit differentially so. In their power to enact relatedness, all needy bodies are the same, yet never identical.

Ultimately, when deprived of articulate subjectivity, it is on this level of political life where oppositions and differences become inscribed and reiterated. When all needy bodies as *différance* enact their relatedness in and with other bodies of *différance*, what emerges is a *différantial* play or struggle over which needs come to matter as relations of care, and which materialize as relations of neglect. In this play of *différance*, new epistemic orders are always in the making, as bodies expose themselves to each other as vulnerable and needy. Here, the body in need as *différance* emerges as "the most general structure of the economy" without which "there is no economy" [62] (p. 9).

*Différance* is always about a constant struggle over the (re)appropriation of value that draws the boundaries of epistemic orders. In the ethics of needs, the needy body that enacts relatedness emerges as such a shifting point of origin for economic and political orders: As *différance,* the bare-body-in-need demanding relatedness with other bodies is the most general structure of the economy or of emergent economies in the making. Without it, there would not be the differentiation camouflaged as sameness that all economies depend upon.

The economies that emerge with relations enacted by needy bodies are always open for the political. This is because the needy body as *différance* obligates a constant rewriting of human relatedness. Or to put it otherwise, the needy body as *différance* writes the world with the relations its needs enact. Writing here, of course, is not to be confused with its literal meaning, but is to be understood in the general sense, where bodies are never simply passive surfaces of inscription, or texts to be written from "outside" by historicized discourses of differentiation. In the general sense of writing, "nature scribbles" and "flesh reads" [45] (p. 127), and in the ethics of needs, it does so because the human body is existentially dependent on care provided by other carnal human beings. While some bodies' needy scripts are readable in the predominant epistemes as demands of care, the scripts for care written by needy others are ~~erased~~ and rendered incomprehensible and ~~unspeakable~~.

It thus becomes possible to see that, when ~~the subject of the speaking, thinking, and caring "I"~~ is written under erasure, the differentially powerful embodied needs make a language of (and for) the world that is not written in words but in corporeal relations of care, as well as neglect, the absence of ~~caring~~. Here, all human bodies are the same at the level of the lowest common denominator, and yet always *différantial* in power. The ethics of needs is a call to deconstruct this kind of body writing, with respect to the subaltern bodies that cannot write the kinds of texts that those response-able can or are "programmed" to read in the prevailing epistemic order.

Epistemic orders and existing governmentalities still matter. Bodies' capacities to write the world with the relations their needs enact does not mean that corporeal relations of care and neglect would just freely float in the air or cut through political voids. Relations and trajectories of care both disrupt and are limited by the changing structures of governance and governmentality. The corporeal relations of care and neglect are also never simply relations between two individuals. They always cut through, and emerge with, a range of biopolitical governmental orders, which aim to manage the very relatedness of life, across populations. These governmentalities operate as grammatical rules by which bodies' writing of relatedness must abide if their writing is to make sense. The various grammars of neoliberal governmentality still issue their subtle instructions on how to relate with the needy, and which needs are to be read as response-able. However, due to the body's capacity to operate as *différance*, these grammatical rules of governmentality are recurrently rewritten and subverted, as needy bodies enact relatedness in unexpected ways. To cite Vicki Kirby, in such a "rhythm of *différance* the body is never not musical. The body is the spacing, the ma(r)king of an uncanny interlude" [45] (p. 63).

In Berenice Fisher and Joan Tronto's well-known definition, care is understood as "*a species activity that includes everything we do to maintain, continue, and repair our 'world' so that we can live in it as well as possible. That world includes our bodies, our selves, and our environment, all of which we seek to interweave in a complex, life-sustaining web*" [18] (p. 40, emphasis in original). The ethics of needs partially describes, and seeks to map,

this very same web of relatedness, albeit from another perspective. Instead of focusing on caring as a "species activity", the focus is on the needs that enact or do not enact care. Then, it is not simply caring that does the weaving of the complex web of inter-connectedness, but also the embodied needs that ask for and demand a response—any response. However, the web that the needs weave is only partially the same as the life-sustaining web of care. Namely, the web that the needs interlace in the world also includes those relations where care needs are responded to with ignorance, neglect, or even violence. This web—or text of the world—exposes transnational (bio)political economies of care, making visible not only those relations where life is sustained but also those where it is not. At the level of the lowest common denominator, it reveals which bodies enact caring relations, and which enact only care that could have been. Furthermore, it exposes which "worlds are being maintained, and at the expenses of which others" [2] (p. 44).

Furthermore, since the ethics of needs writes ~~(human) subjectivity~~ under erasure, there is no need to limit the theory to apply only to the human species. Nonhuman bodies also enact relations of care, as well as relations of neglect and violence in this world, and today it is perhaps more urgent than ever to try and read those relations, their entanglements, and the hierarchies therein between relations of care and neglect. The ethics of needs is therefore about a politics of care beyond species activity, which could perhaps be utilized in developing a more planetary ethics of needs.

But how does the ethics of needs challenge the neoliberal politics of life? I have discussed how the incapacity of neoliberal biopolitics to engage with dying, decaying, and "incapacitated" life makes it a politics of life, where life is given care and made to live only insofar as the living body is either economically productive through its labor, or profitable through the sales or financialization of its needs. Beyond this, life may as well be let to die or killed without sacrifice, as far as neoliberal governmental reason is concerned. Such violence against economically non-productive bodies is immoral. Yet, in the prevailing epistemic order, there are far too many examples where we can see this rationality at play, ranging from the politics of eldercare to global vaccination politics, to the global distribution of human resources for health to the instrumentalization of non-human life in capitalist modes of production. For some reason, such a neoliberal politics of life seems very difficult to resist and subvert, regardless of its implicit (and often explicit) violence towards the presumably speechless bodies of the needy.

Here, the question remains as to whether a kind of transnational politics of life is possible, where care needs would be seen as valuable and capable of enacting caring responses also when the care involved would not feed into transnational capital accumulation. In lieu of a conclusion, I want to suggest that the ethics of needs provides an opening for a more ethical biopolitics. To the extent that neoliberal biopolitics is about a politics of making live and letting die, its moral problem derives from the corporeal ontology of care, and the material resources it requires: Making live and letting die involves decisions over which lives are worthy of life-sustaining care, and which can be left to die. After all, the ontological condition of care is that it is always a concrete relation with a particular body-in-need. Therefore, when taking care of somebody—or some population, as in nationalized vaccine politics for instance—it is an ontological necessity for the caring body or institution to turn its back on the needs of some others. This means that one may *care about* a whole variety of corporeal care needs in the world, but for no embodied subject, population, or institutional construct is it ontologically possible to concretely *take care of* and *provide care* for everybody at the same time [18]. In all care relations, some needs and some bodies remain neglected.

In the ethics of needs, corporeal relations of care and neglect are understood as being deeply entangled, and these entanglements emerge continuously and everywhere, through the choices and decisions as to whose needs count for the response-able. While the "bare" body-in-need, living in the field of *zoē*, is generally silenced in the neoliberal episteme, the ethics of needs allows us to see how within this allegedly non-political sphere of life, there are still relations of power. Some needy bodies, even at the level of the lowest common denominator, dominate others, and this is because the *needs* of some bodies are economically

and politically more powerful, and hence more worthy, and speakable compared to the utterly muted bodies of Others. When care is defined as a corporeal-relation-enacted-by-needs, these material-discursive *hierarchies of needs become a property of care*. This means that the ~~care that is absented~~ to make care present elsewhere is part and parcel of the corporeal relations of care. Neglect is thus ~~care~~ that could have been. When analyzing the political relatedness that caring enacts in the world, it is perhaps a moral necessity of the care ethicist to also try and address the political relations of ~~caring~~ that are absented when care is provided. Such a perspective makes care (including the care of populations) always a potential question of neglect and/or violence.

When striving for a more caring politics of life than neoliberal biopolitics, we must remember that regardless of all the possible care in the world, eventually, all living bodies become incapacitated and die. Indeed, if its sole goal was to sustain life, care would be forever doomed to fail. But then, care is not only about the sustenance of *life* or economically productive life, even if neoliberal biopolitics may be directed to this end. Rather, care is about responding to the needs of others, and about corporeal relations where *needs rather than life itself* count as originary starting points for politics. Consequently, *the biopolitics of needs* that emerges with the ethics of needs is not just about the politics of *life*. The biopolitics of needs is, literally, about the *politics of life of needs*. In such biopolitics, it is recognized that needs have a life of their own, even when the life of a speaking, caring, and economically productive subject seizes to exist. In the politics of life of needs, it becomes possible to address the care needs of the dying and the incapacitated, also when the care provided fails to accumulate capital, or be otherwise economically productive in the capitalist sense of productivity.

In other words, for care to be an ethical relation, we must let the needs of Others haunt the epistemic orders and governmental relations that are understood as presumably caring. The potentially more caring biopolitics that emerges with the ethics of needs requires a recognition of the difficult choices that care, as a response to the *living needs* of others, always demands. It is an approach that necessitates a constant deconstruction of care relations. The ethics of needs always returns to those to whom our backs were turned when care was provided, and to the excess of life that the neoliberal biopolitics fails to account for. It returns to the impossible limits of ~~care~~—recurrently, persistently, chronically.

**Funding:** This research was funded by The Academy of Finland, grant number 321972, and The Kone Foundation, grant number 201802636.

**Acknowledgments:** I want to thank the editors of this special issue, Maggie FitzGerald and Maurice Hamington, for inviting me to participate in the issue, and thereby providing the opportunity to finally try and put this argument together in an article format. Thank you for your patience with my delays and requests for deadline extensions. Heartfelt thanks also go to the two anonymous reviewers for their encouraging and thought-provoking comments on the first draft of the manuscript. I think I have failed to fully respond to all your critiques, but I promise to go back to them when developing my ethics of needs further in the future. It is certainly still a theory that is underdeveloped and full of gaps, but thanks to your caring labor of reading and commenting, this first version of the argument is now "out there". The article is based on my PhD Thesis *The Global Biopolitical Economy of Needs: Transnational Entanglements between Ageing Finland and the Global Nurse Reserve of the Philippines* (University of Tampere, 2017). While this article is a theoretical piece of writing, I remain indebted for my research participants for helping me to think through the ethics of needs empirically, as transnational phenomena. Thank you also for the examiners of the thesis, Eeva Jokinen, Fiona Robinson, and Anna Agathangelou for all your feedback and encouragement at the time when the theory was first developed. Anna "Imppu" Rajala, dear friend and colleague: Thanks for reading this thing through and just being the wonderful colleague and person you are. Sanna Melonen and Sini Rönnqvist: Thank you for all the cooking and feeding and care you provided me with during our week in Levi, Lapland, during the writing-retreat-with-some-skiing, when I finished the first and the most difficult draft of this manuscript. I needed the get-away, the food, and your company, and it was all just lovely.

**Conflicts of Interest:** The authors declare no conflict of interest.

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
