# Peer review of "An Ethics of Needs: Deconstructing Neoliberal Biopolitics and Care Ethics with Derrida and Spivak"

_philosophies, doi:10.3390/philosophies7040073_

Round 1

Reviewer 1 Report

I thoroughly enjoyed reading this thoroughly interesting paper, which I felt presents a compelling critique of care ethics and offers a productive way to develop care ethical analyses towards an ethics of needs. I offer general comments, below, in the spirit of appreciative curiosity about the details of the argument offered in the article, followed by 2 specific recommendations. The latter are intended to reflect my experience of reading the paper for the first time, as they relate to the order in which ideas are introduced. Finally I mention the typos I noticed.

General comments

  • I wonder if referring to specific infrastructural or state-provision of medical or other services, e.g., 'financialised care-providers' (line 63) with the word 'care', might be to fudge the distinction, which later becomes important, between care of caring subjects vs care as bare needs.

  • Is there a tension in the shift towards homo egenus (as first articulated at lines 172-198) may be in tension with the foundational versions of care ethics which focus on the care-giver in part due to the (admittedly narrow) political implications of erasing women's labour in the home. If the patriarch, husband, or the elite, sucks up caring resources, might that confuse or complicate our ability to recognise the 'body in need of care'?

  • Line 459-460 - is this the same as saying that (classic versions of) care ethics is/are implicitly ableist? If so, I wonder why not further emphasise this dimension of the critique?

  • The consideration of more-than-human worlds and lives is apt. But I think the implications are wider-ranging than currently reflected in the article. If 'our barest needs' defined by 'lowest common denominator of embodiment' are productive of relationships in the sense articulated at lines 605-611, does this not imply that nonhuman animals (at least the ones who are capable of responding to one another's needs with care, such as great apes, elephants, cetaceans) are acting unethically in the same way as humans when they turn their backs on others of their kind who have needs for care? Are non-human animals (as speechless needy bodies) always stuck in the position of subaltern in this ethical theory? Does it matter if we cannot be sure that these species have an equivalent to the 'caring I' apparent in human care relations? I think this might be related to the definition of 'species activity' with which you take issue in Fisher and Tronto's understanding of care.

  • Are there 2 senses of 'death' in use towards the end of the paper? Why is 'respect' of/for death (line 892) an ethical appropriate response to the prevailing neoliberal biopolitics of 'letting die' (lines 897-898)? It may be that this is answered implicitly at lines 923-933, but I think it needs more explanation. What would be the difference between respecting death, perhaps 'caringly' (?), as opposed to allowing to die as a result of the neoliberal refusal to provide care?

Specific recommendations

  1. I recommend that the consideration of the author's complicity/embeddedness in the critique they offer in lines 499-505 be added to the introduction to the paper. As a reader I was looking for this kind of reflection and felt that introducing it half-way through the paper was too late.

  2. Similarly, in view of the attention paid to critical disability studies at points throughout the paper, I think this should be mentioned earlier. By the time I had read up to line 130, I was wondering 'where are critical disability studies in this argument?' so it would seem sensible to mention this angle forms part of the analysis presented.

Typos

  • Line 8 - Jacques
  • Line 103 - the sentence is just a bit unclear. I think changing "with subjects" to "to subjects" would help or maybe there is a word missing.
  • Line 162 - Jacques
  • Line 518 - 'quite cannot speak' didn't read well for me --> should this be 'cannot quite speak' ?
  • Line 536 - in Spivak block quote, should this be "Self's shadow" ?
  • Line 869 - is there a word missing? I don't understand "and which only care that could have been"?

Reviewer 2 Report

I have enjoyed reading this stimulating, original, and well-researched essay.  Thoughtful and provocative, the article makes a good case for: 1) the relevance of Derrida and Spivak’s writings for care ethics; 2) the desirability of assigning far greater significance to the “body in need” within care ethics (as opposed to (an excessive focus on) what the author labels “caring practices”). 

For the first claim, however, I felt at times that the title ‘promised’ a little bit more than the paper delivered: wasn’t there a lot more Foucault than Spivak at the end of the day?   I am not asking the author to change her/his title; I merely wish to underscore the fact that Spivak was perhaps given too small a space (some of her shorter essays and interviews (e.g. “The Post-colonial critic”, “The Problem of Cultural Self-representation” could have been given more room in my view, had there been sufficient space).  Nevertheless, as mentioned above, I do think that the article offers an original contribution to care ethics by bringing Derrida and Spivak into this sub-field and by making a rich and multilayered argument for an “ethics of needs”.

Regarding the second claim: I was not entirely convinced (although I remain open to being convinced!) that there is that much care ethics scholarship out there that does not attach significance to both “bodies in need” (as understood by the author) and to “caring practices”.  (But perhaps this is because I failed to fully grasp the radicality of what was proposed about needy bodies?)  Either way, I think it might helpful for the author to cite a few specific examples of texts and authors he/she has in mind when he/she speaks of the “traditional accounts of the ethics of care” (ex: lines 165-167).  Who are we talking about here?  This should be made clearer (and this is one of my main suggestions for ‘substantive’ revisions).  It seems quite important to give readers a better sense of which texts are referred to by the author, especially given the significant charge levelled in this article (i.e. that traditional accounts of care ethics partake in epistemic violence and mute the subaltern needy body).   

That being said, I should say that I agree with most of the concerns and criticisms levelled in this piece – e.g. the fact that many care ethicists do not sufficiently consider violence and neglect. This is a very important critique and corrective.   (e.g. “Neglect is a species activity, too.”  Yes! Nicely put!  (line 603))

A small observation:  while it is ok if the author does not wish to engage with Sarah C. Miller’s work, I am not convinced that the argument given for not doing so is a very solid one (i.e. that their philosophical premises are different and in tension).  (Sometimes, can’t it be highly fruitful to engage precisely with authors who start with radically different premises?)  More generally, I wonder whether it could have been helpful to more explicitly and concretely acknowledge a few care ethicists’ accounts of needs (several care ethicists, for instance, have drawn on Nancy Fraser’s rich analysis of “the politics of needs interpretation”… Could Fraser have something helpful to say here?). Given the fact that the article is already very long and already accomplishes so much, I do not wish to insist that the author address this criticism extensively in his/her revisions.  This is mostly “food for thought”.      

Minor revisions to do please.

It might be good to reformulate or fix the following:

  • Jacques Derrida’s name is not correctly spelled in some places, including the abstract (the “s” missing). Please check.
  • Lines 47-48 - “care provision and access to adequate care are increasingly dependent on the worth of the care-receiver to the accumulation of transnational capital” (the claim is not very clear to me)
  • Line 80 – “I argue that the care ethicists’ focus…”. (here, “the” probably could be removed)
  • Line 103 “the care ethics that emphasizes” (remove “the”?)
  • Line 117 “some elaborations and delineations are at place” (a bit imprecise? Could you reformulate or say a bit more perhaps?)
  • Lines 154-56: “The ethics of needs goes beyond care – or does not necessarily reach the event where caring takes place.” (Please clarify – I am not sure that I follow here)
  • Lines 690-91: “where the moral power of care is present only its own absence.” (not sure I understand – perhaps a word is just missing?)
  • Line 658 - “to develop ways read the relatedness” (word missing here?)
  • Line 858 – “emphasis in original” (I don’t see the emphasis)

And finally -- two modest suggestions for later work (not necessarily this piece!).  In several places in the article, I thought that bringing in some literature on necropolitics (especially those pieces directly inspired by Foucault and Mbembe) might be quite pertinent for some of the issues tackled.  Also, one care ethics book chapter that might be pertinent to consider down the road (a chapter that draws on Agamben quite heavily):  Monique Lanoix, “Barefoot, Pregnant and in the Kitchen: Critical Reflections on the Domestic through the Lens of Bare Life” (2019), in In Yet a Different Voice: Care ethics and the francophone contribution.

Many thanks again for this stimulating article.
